# Membrane-Supported Layered Coordination Polymer as an Advanced Sustainable Catalyst for Desulfurization

**DOI:** 10.3390/molecules26092404

**Published:** 2021-04-21

**Authors:** Fátima Mirante, Ricardo F. Mendes, Rui G. Faria, Luís Cunha-Silva, Filipe A. Almeida Paz, Salete S. Balula

**Affiliations:** 1REQUIMTE/LAQV & Department of Chemistry and Biochemistry, Faculty of Sciences, University of Porto, 4169-007 Porto, Portugal; fatimaisabelmirante@gmail.com (F.M.); up201202396@fc.up.pt (R.G.F.); l.cunha.silva@fc.up.pt (L.C.-S.); 2CICECO—Aveiro Institute of Materials, Department of Chemistry, University of Aveiro, Campus University of Aveiro, 3810-193 Aveiro, Portugal

**Keywords:** catalytic membrane, layered coordination polymer, oxidative desulfurization, hydrogen peroxide, lanthanides

## Abstract

The application of a catalytic membrane in the oxidative desulfurization of a multicomponent model diesel formed by most refractory sulfur compounds present in fuel is reported here for the first time. The catalytic membrane was prepared by the impregnation of the active lamellar [Gd(H_4_nmp)(H_2_O)_2_]Cl·2H_2_O (UAV-59) coordination polymer (CP) into a polymethyl methacrylate (PMMA, acrylic glass) supporting membrane. The use of the catalytic membrane in the liquid–liquid system instead of a powder catalyst arises as an enormous advantage associated with the facility of catalyst handling while avoiding catalyst mass loss. The optimization of various parameters allowed to achieve a near complete desulfurization after 3 h under sustainable conditions, i.e., using an aqueous H_2_O_2_ as oxidant and an ionic liquid as extraction solvent ([BMIM]PF_6_, 1:0.5 ratio diesel:[BMIM]PF_6_). The performance of the catalytic membrane and of the powdered UAV-59 catalyst was comparable, with the advantage that the former could be recycled successfully for a higher number of desulfurization cycles without the need of washing and drying procedures between reaction cycles, turning the catalytic membrane process more cost-efficient and suitable for future industrial application.

## 1. Introduction

In spite of recent strides in the development of alternative energy sources, the contribution of fossil fuels to global energy demand still remains at about 80% [1,2]. The burning of these fuels leads to the concomitant emission of different pollutants, such as carbon monoxide (CO) and sulfur or nitrogen oxides (SO_x_, NO_x_) [3,4]. Sulfur oxides in particular tend to react with water in the atmosphere, producing harmful acids which have several adverse effects on human health and the environment [5]. This has led to a strict worldwide goal of limiting sulfur content in fuels [6]. Currently, hydrotreating processes are the preferred industrial method for the removal of heterocyclic sulfur molecules from fuels, having a remarkable efficiency in the removal of compounds such as mercaptans and thiophene [7]. Hydrodesulfurization is not, however, as effective for the removal of heavier sulfur-containing compounds such as dibenzothiophene (DBT) and its derivatives [8,9]. To achieve high desulfurization efficiency by this process, extreme reaction conditions are required (temperatures >350 °C and pressures in the 20–130 atm H_2_ range) [10,11]. Alternative/complementary desulfurization methods have emerged as more cost-effective and sustainable approaches [12,13,14]. Among these, catalytic oxidative desulfurization (ODS) is particularly interesting, as it can be employed in tandem with liquid–liquid extraction or adsorption processes, ensuring removal of refractory sulfur compounds from fuels under mild temperatures and atmospheric pressure [15]. The combination of extraction and catalytic oxidative desulfurization (ECODS) was demonstrated to be a highly efficient method for the production of sulfur-free fuels [16,17,18,19]. The oxidation of sulfur compounds in ODS processes is usually carried out using hydrogen peroxide as the oxidant, an environmentally benign oxidant with highly active oxygen content (47%) [15]. The preparation of adequate ODS catalysts remains one of the main quests of researchers in this field, with the requirements of high activity and ease of handling during recovery from the reaction medium and during the recycling process.

Coordination Polymers (CPs) and Metal-Organic Frameworks (MOFs) are a class of organic–inorganic hybrid materials formed by metallic centers (clusters or ions) coordinated by organic bridging linkers, which have gained attention in several fields of research because of their unique properties such as high porosity, tunable structure and functionality, extensive surface areas, and remarkable thermal, chemical, and mechanical stabilities [20]. Even though the main use of CPs/MOFs in catalysis has been as host materials for homogeneous catalysts [21,22,23,24,25], some of these materials exhibit intrinsic catalytic activity for ODS [26,27,28,29,30,31]. We have recently demonstrated that a positively charged, robust, and dense lamellar CP based on the combination of a triphosphonic acid linker [nitrilotri(methylphosphonic acid), H_6_nmp] and a rare-earth metal cation, Gd^3+^ (formulated as [Gd(H_4_nmp)(H_2_O)_2_]Cl·2H_2_O and coined hereafter as UAV-59) [32], is capable of catalyzing the complete removal of sulfur compounds from a model fuel comprised of 2000 ppm of highly refractory sulfur compounds, using an ECODS system under environmentally friendly conditions [33]. Applicability in the industry is the main goal in the development of ODS catalysts, and CPs/MOFs still have not found their footing for industrial application as they suffer from the inconvenience of being powdered solids, which not only hinders their ease of use, but can lead to deactivation via agglomeration or sintering, ultimately reducing the number of available active sites. Recently, membrane-supported CPs/MOFs have been reported, but their applications have mostly been centered on gas separation [34,35,36,37].

In the present manuscript, we report the first use of a catalytic membrane in oxidative desulfurization process. Polymethyl methacrylate (PMMA, acrylic glass), an organic polymer, was employed as a supporting membrane for the immobilization of UAV-59, with the intent of mitigating the easier handling of the powdered catalyst and to avoid its deactivation. The PMMA membrane is employed for the first time in desulfurization experiments because previously only molecular dynamic simulations were reported [38]. This membrane was chosen since it is catalytically inert, which allow for a better comparison of UAV-59 in powder with its supported membrane. Also, PMMA is one of the most used polymers in different industries, being thermally stable at the conditions used in this work, as well as being light weight and resistant to impact. The prepared UAV-59@PMMA catalytic membrane was tested as a catalyst in ECODS process to treat a multicomponent fuel under sustainable conditions. The stability and recyclability of the catalytic membrane was evaluated.

## 2. Results and Discussion

### 2.1. Membrane Preparation and Characterization

UAV-59 was supported into a PMMA matrix membrane using a simple casting method as depicted in Figure 1. The CP in the powdered form was suspended in a PMMA solution in dichloromethane and cast into a mold. After solvent evaporation at ambient temperature the UAV-59@PMMA membrane was isolated and characterized.

Membrane preparation was optimized by testing different PMMA-to-UAV-59 ratios and quantities. Higher UAV-59 quantities led to a lower mechanical stability with the membranes breaking more easily. This limitation can be surpassed by maintaining the amount of PMMA similar to or greater than UAV-59, with the optimal ratio being 1:1 (*w/w*). At this ratio, the amount of material used in the membrane preparation does not influence its mechanical stability. Membrane homogeneity is however affected at lower ratios with small pockets of material being observed in the membrane. For these reasons, the optimal membrane conditions used in this work were a 1:1 ratio prepared with 0.6 g of each component.

The incorporation success of UAV-59 into the PMMA matrix was accessed by powder X-ray diffraction and SEM/EDS analysis. Incorporation is immediately visible to the naked eye because of clear visual modification of the membrane aspect: pure PMMA membranes are transparent, while the CP-supported membrane has a solid white coloration (the same color as for UAV-59 crystals; Appendix A). UAV-59 crystallinity was not hindered during the membrane preparation, with the material maintaining its structural crystalline features (Figure 2). Regarding homogeneity, as referred before, though because of the typical large crystal size of UAV-59 small pockets of material may appear during the membrane preparation. The optimized conditions used for membrane preparation allowed UAV-59 to be uniformly dispersed as depicted in Figure 3. EDS mapping of the heavy elements present in UAV-59 show an unequivocal uniform distribution of the material throughout the membrane.

### 2.2. Desulfurization of a Model Fuel

We reported recently the high catalytic efficiency of the powdered layered coordination polymer UAV-59 for the oxidative desulfurization of a multicomponent model fuel [33]. Even though the heterogeneous catalyst showed high performance, inherent mass loss and extra laborious removing techniques (centrifugation or filtration) for catalyst re-use are always associated at the end of the process or even during the recycling study. To avoid these limitations, we found it to be crucial to convert the solid powder into a catalyst that could be more easily handled and re-used. We achieved this by the impregnating the powdered UAV-59 into a PMMA membrane matrix, resulting in a new catalyst composite that was easier to add and remove from the desulfurization system.

The catalytic performance of UAV-59@PMMA membrane was investigated for the desulfurization of a model diesel containing four refractory sulfur compounds most commonly found in liquid fuels, namely 1-benzothiophene (1-BT), dibenzothiophene (DBT), 4-methyldibenzothiophene (4-MDBT), and 4,6-dimehtyldibenzothiophene (4,6-DMDBT). The desulfurization process was performed in a biphasic system composed of the model diesel and [BMIM]PF_6_ employed as extraction solvent and using H_2_O_2_ as oxidant. The desulfurization occurred in two main steps: (i) an initial extractive desulfurization performed by vigorously stirring for 10 min at 70 °C, where sulfur compounds are transferred from diesel to the extraction solvent and (ii) oxidative desulfurization, started by the addition of the oxidant, where the sulfur compounds mainly present in the extraction solvent phase are oxidized to the corresponding sulfones and/or sulfoxides. The oxidation of the sulfur compounds occurred by the activation of the oxidant by the catalyst, i.e., the UAV-59@PMMA membrane. This contrasts our previous studies which solely employed the powdered catalyst which was spread among both the diesel and extraction solvent phases. In the present work, the catalytic membrane was incorporated in a static reactor compartment while in complete contact with the biphasic diesel/[BMIM]PF_6_ system (Figure 4).

#### 2.2.1. ECODS Process: Efficiency and Sustainability

An optimization study was performed to make the process more efficient, sustainable, and cost effective. Three main parameters were investigated: number of catalytic centers (Gd^3+^ content in the membrane), oxidant amount, and the ratio of model Fuel/[BMIM]PF_6_ solvent. The catalytic performance of the UAV-59@PMMA membrane was compared to the powdered catalyst UAV-59 (prepared by a one-pot procedure, denominated as 1op in reference [33]), using the same amount of Gd^3+^ active centers (0.04 mmol). Figure 5 depicts the results obtained, where it is possible to observe that after 4 h a near complete desulfurization was achieved using the powdered materials and the catalytic membrane. Using the powdered catalyst, the oxidative desulfurization is initiated practically after the initial extraction step, in opposition to the catalytic membrane where an induction period is observed between 1 and 3 h of reaction. Another important aspect is the superior initial extractive desulfurization (before the addition of oxidant) observed in the presence of a fixed catalytic membrane (an increase by c.a. 10%) when compared to the powdered catalyst. It is therefore crucial to improve the oxidative desulfurization efficiency of ECODS systems by employing the catalytic membrane via optimization of the reaction parameters.

The size of the catalytic membrane, i.e., the amount of active catalytic centers (quantified as Gd^3+^ centers) present in the ECODS system was investigated by using half the amount of that previously used in the comparative study with the powdered catalyst. Figure 6 summarizes the desulfurization profile using 0.04 and 0.02 mmol of Gd^3+^ showing a similar catalytic performance during the induction period, observed between 1 and 3 h of reaction, with a near-complete desulfurization achieved just after 4 h. Experiments described hereafter were, therefore, performed using UAV-59@PMMA membrane incorporating only 0.02 mmol of Gd^3+^.

The oxidant amount is an important factor because, alongside with the catalyst itself, it has a significant influence on the efficiency of the oxidative desulfurization step by way of oxygen donation. Three different oxidant amounts were evaluated (100, 75, and 50 µL, corresponding to 0.86, 0.64, and 0.43 mmol, respectively). The desulfurization results obtained using [BMIM]PF_6_ as extraction solvent at 70 °C are summarized in Figure 7. Similar desulfurization profiles were observed between ECODS processes using 0.86 and 0.64 mmol of H_2_O_2_, attaining a complete desulfurization of the model diesel after 4 h. The same induction period was observed for up to 3 h of reaction using different oxidant amounts. In the presence of a less quantity of H_2_O_2_, the extension of the oxidative desulfurization was smaller, and the complete desulfurization could not be achieved under the 4 h time frame observed for the other conditions. We infer, therefore, that an optimal H_2_O_2_ amount could be considered to be 0.64 mmol.

Another important parameter that can have a remarkable influence in the ECODS system is the ratio of the model diesel:[BMIM]PF_6_ solvent. The use of smaller amounts of ionic liquids turns the ECODS process more sustainable and with higher economic viability. Different ECODS systems were studied using 1:1 and 1:0.5 model diesel:[BMIM]PF_6_ ratios, and under an extraction solvent-free system (Figure 8). The initial extractive desulfurization between 1:1 and 1:0.5 was similar (around 40%). A remarkable difference in the oxidative desulfurization profile was, however, observed between these ECODS systems. This behavior was already reported before using the powdered UAV-59 catalyst [33] and it is additionally observed when the same layered catalyst is incorporated into the PMMA membrane. Using the 1:0.5 diesel:[BMIM]PF_6_ system, the concentration of sulfur compounds present in the extraction phase was double compared to the 1:1 system, since sulfur initial extraction was similar. Therefore, a higher contact between these and the H_2_O_2_ oxidant must be higher, what may promote the catalytic oxidative reaction, increasing desulfurization efficiency (94% of desulfurization was registered after 3 h using the 1:0.5 system, instead of 65% achieved with 1:1). The resulting percentage of desulfurization obtained without extraction solvent being used was 21% desulfurization after 4 h of reaction, which indicates that the oxidative catalytic reaction may mostly occur in the [BMIM]PF_6_ phase (Figure 8). Without the presence of the catalyst in the ECODS process (Figure 8) or the PMMA membrane without the layered UAV-59 polymeric catalyst (not shown), no oxidative desulfurization occurred and the extraction of sulfur from model diesel is not increased after the initial extraction step.

#### 2.2.2. Membrane Recycling

Recycling of the UAV-59@PMMA catalyst was investigated by using the same piece of composite membrane in various consecutive ECODS cycles. For all cycles the optimized ECODS conditions were maintained (1:0.5 model diesel:[BMIM]PF_6_, 0.64 mmol of oxidant, membrane containing 0.02 mmol of Gd^3+^, 70 °C). During the recycling process two different procedures were performed: (i) the membrane was separated from the reaction medium after each cycle and washed with acetonitrile and water and then dried in a desiccator and (ii) the membrane was removed from the ECODS system and used directly in the next cycle without any treatment.

Figure 9 summarizes the results obtained using both procedures showing that the washing practice between cycles leads to partial deactivation of the membrane: the catalytic performance decreases immediately from the first to subsequent cycles. It was noteworthy that using the catalytic membrane directly between cycles promotes catalytic stability and the concomitant near complete desulfurization for at least six consecutive ECODS cycles. This behavior must be related to the high permeability of this membrane, mainly after its washing with acetonitrile and water, which causes a loss of stiffness and partial winding that probably can contribute to a diffusion decrease of compounds through the membrane, and a lower accessibility to the active catalytic centers (see next section for further details). The usage of extra solvents should, in this way, be avoided during ECODS processes. Moreover, from a technological viewpoint, this is an important advantage for any future and potential industrial application.

### 2.3. Membrane Stability

The membrane stability was accessed in different conditions: temperature and solvent immersion. The prepared membrane is stable up to 150 °C without modifications on the structural integrity of the coordination polymer. While the membrane shows signs of a decrease in mechanical stability (i.e., more easily broken) the overall stability is maintained (Appendix A). The solvent in which UAV-59@PMMA is immersed influences the membrane stability. For this, UA-59@PMMA was immersed for 24 h in 5 mL of water, acetone, ethanol, methanol, hexane, acetonitrile, or toluene. In general, the structural integrity of the membrane and the structure of the supported CP remain unaltered (Appendix A) except when immersed in acetone, toluene, or acetonitrile. UAV-59@PMMA completely dissolves in acetone after only 15 min, and the membrane integrity greatly decreases in the presence of toluene and acetonitrile. Although the membrane is not completely dissolved in the presence of these two solvents, a partial dissolution occurs with the membrane resembling a wet piece of paper.

These studies help to better understand the recyclability studies described above. As observed in Figure 9, the membrane catalytic activity decreases after the first cycle when the membrane is washed. Because the washing process was performed with acetonitrile (which permits a much better removal of the chemical components of the catalytic system) the membrane stability was compromised, leading to both a decrease in mechanical stability and to a partial dissolution. It is, thus, necessary to optimize this step in the near future to employ a friendlier solvent in the case for the need to wash the catalytic membrane.

The same structural change of the CP previously reported by us, when the catalytic studies were performed with UAV-59 in the powdered form [33] were observed for the material supported in the membrane, with this being the active catalytic species. In the present work, the same transformation is visible once again in the first catalytic cycle (Appendix A). This transformation is not, however, complete during the first cycle (most likely due to the PMMA barrier), occurring during the second one. Despite this transformation, the material crystallinity and structural features remain unchanged during the remaining catalytic cycles (Figure 10). Membrane stability and homogeneity are maintained as well during the performed studies. As depicted in Figure 11, the transformed material maintains its homogeneous distribution at the surface and throughout the membrane. The sole visible modification concerns the crystal morphology the catalytic reaction tends to break apart the crystallites with the plate-like morphology of the parent material being modified to a more rounded and uncharacteristic shape, in line with that previously observed for the powdered form of the catalyst [33].

## 3. Experimental Section

### 3.1. Materials and Methods

The following chemicals and reagents were purchased from Sigma-Aldrich (unless otherwise indicated) and used as received: 1-benzothiophere (98%), dibenzothiophene (98%), 4-methyldibenzothiophene (96%), 4,6-dimethyldibenzothiophene (97%), *n*-octane (98%), 1-butyl-3-methylimidazolium hexafluorophosphate (97%), tetradecane (99%), and 30 wt.% hydrogen peroxide; gadolinium(III) oxide (at least 99.99%, Jinan Henghua Sci. & Tec. Co. Ltd., Jinan, China); nitrilotris(methylenephosphonic acid) [H_6_nmp, N(CH_2_PO_3_H_2_)_3_, 97%, Fluka]; hydrochloric acid (HCl, 37% Analytical Reagent Grade, Fisher Chemical), and Methyl Methacrylate Polymer (TCI).

Inductively Coupled Plasma—Optical Emission Spectrometry (ICP-OES) analyses were registered with a spectrometer Optima 4300 DV (Perkin Elmer, Waltham, MA, USA) with a plasma source (RF generator of 40 Hz) and automatic sampler (PerkinElmer AS93-plus). ICP-OES analyses were performed at “Centro de Apoyo Científico-Tecnológico (CACTUS) de la Universidad de Santiago de Compostela, USC (Galicia, Spain)”. SEM (Scanning Electron Microscopy) images were acquired using either a high-resolution Hitachi SU-70 working at 4 kV, or performed in a FEI Quanta 400 FEG ESEM high-resolution scanning electron microscope equipped with an EDAX Genesis X4M spectrometer working at 15 kV. Samples were prepared by deposition on aluminum sample holders followed by carbon coating using an Emitech K950X carbon evaporator or coated with an Au/Pd thin film by sputtering using an SPI Module Sputter Coater equipment. Routine Powder X-Ray Diffraction (PXRD) data for all materials were collected at ambient temperature on a Empyrean PANalytical diffractometer (Cu Kα_1,2_ X-radiation, λ_1_ = 1.540598 Å; λ_2_ = 1.544426 Å), equipped with an PIXcel 1D detector in transmission mode in a Bragg–Brentano para-focusing optics configuration (45 kV, 40 mA). Intensity data were collected by the step-counting method (step 0.04°), in continuous mode, in the c.a. 5 ≤ 2 θ ≤ 50° range. Catalytic reactions were periodically monitored by GC-FID analysis carried out in a Bruker 430-GC-FID chromatograph (Bruker Daltonik GmbH, Bremen, Germany). Hydrogen was used as carrier gas (55 cm·s^−1^) and fused silica Supelco capillary columns SPB-5 (30 m × 0.25 mm i.d.; 25-μm film thickness) were used.

### 3.2. UAV-59 and UAV-59@PMMA Membrane Preparation

The preparation and scale-up of [Gd(H_4_nmp)(H_2_O)_2_]Cl·2H_2_O (UAV-59) was done as previously reported [33].

For the membrane preparation, c.a. 600 mg of PMMA was dissolved in 5 mL of dichloromethane (Note: slow addition of PMMA is needed to prevent agglomeration). UAV-59 was added to the solution and was stirred for 20 min. UAV-59 was slightly grinded prior to the addition to PMMA to separate some crystals agglomeration and increase the material dispersion. The resulting suspension was cast to a petri dish (c.a. 5-cm diameter) and stirred until a more consistent suspension was present, and then it was left unstirred overnight. The resulting UAV-59@PMMA membrane was removed from the petri dish and washed with copious amounts of distilled water.

### 3.3. Extractive and Oxidative Desulfurization Process (ECODS)

Desulfurization studies were performed using a model diesel containing a total sulfur concentration of 2000 ppm composed of equal amounts of refractory sulfur compounds, namely: 1-benzothiophene (1-BT), dibenzothiophene (DBT), 4-methyldibenzothiophene (4-MDBT), and 4,6-dimehtyldibenzothiophene (4,6-DMDBT) in *n*-octane. These experiments were carried out under air (atmospheric pressure) in a closed borosilicate 5-mL reaction vessel, equipped with a magnetic stirring bar and immersed in a paraffin bath heated to 70 °C. Processes were performed in a biphasic system composed by the ionic liquid (1-butyl-3-methylimidazolium hexafluorophosphate, [BMIM]PF_6_), and the multicomponent model diesel in the presence of a membrane used as the catalyst and H_2_O_2_ (30% wt. aq.) as oxidant. In a typical experiment, 0.75 mL of model diesel and the extraction solvent (0.75 and 0.38 mL) were added to the catalytic membrane (containing 0.02 and 0.04 mmol of Gd). An initial extraction of sulfur compounds from model diesel to the ionic liquid phase occurred by only stirring both immiscible phases for 10 min at 70 °C. The oxidative catalytic step of the process is then initiated by the addition of H_2_O_2_ oxidant (0.86, 0.64, and 0.43 mmol). The sulfur content in the model diesel phase was periodically quantified by GC analysis using tetradecane as standard. The recycling capacity of the heterogeneous catalyst was investigated using the same portion of membrane in consecutive desulfurization cycles. New portions of model diesel, oxidant, and extraction solvent were added to the used membrane at the end of each desulfurization cycle. Between cycles, the membrane was washed with ethanol. All recycling cycles were performed under the same initial experimental conditions.

## 4. Conclusions

This manuscript reports for the first time the application of a catalytic membrane for oxidative desulfurization process using a multicomponent model diesel. This catalytic membrane was prepared by using a simple casting approach using the catalytically active coordination polymer (CP) [Gd(H_4_nmp)(H_2_O)_2_]Cl·2H_2_O (UAV-59) in a polymethyl methacrylate (PMMA) support membrane. This preparative method allowed for the preparation of a homogeneous membrane UAV-59@PMMA, with UAV-59 being dispersed in a uniform way throughout the membrane. After the optimization study, a complete desulfurization was achieved after 2 h using the ratio of 1:0.5 diesel/extraction solvent ([BMIM]PF_6_), H_2_O_2_ as oxidant, at 70 °C. The powdered catalyst under similar reaction conditions achieved near complete desulfurization after 1 h. However, the use of a one-piece membrane catalyst presents a large advantage avoiding catalyst mass loss and also the absence of extra laborious procedures in catalyst recovery. The recycle capacity of the catalytic membrane was confirmed for six consecutive ECODS without loss of activity. A substantial advantage of using catalytic membrane instead of powdered catalyst was the facility of handling the catalyst between cycles without mass loss during consecutive reaction cycles and without need for catalyst recovery and post-catalytic use treatment (such as washing and drying). Therefore, the successful application of catalytic membrane in ECODS process will increase the feasibility of this technology in an industrial environment.

## Figures and Tables

**Figure 1 molecules-26-02404-f001:**
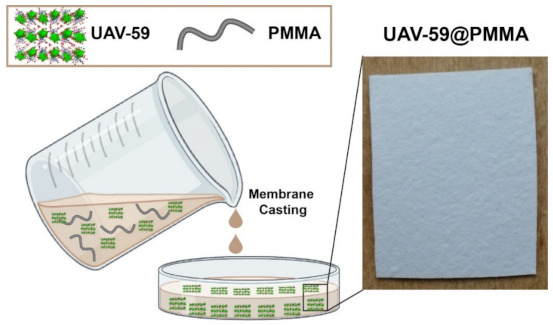
Schematic representation for the preparation of a membrane and a photograph of the UAV-59@PMMA membrane used in the current studies.

**Figure 2 molecules-26-02404-f002:**
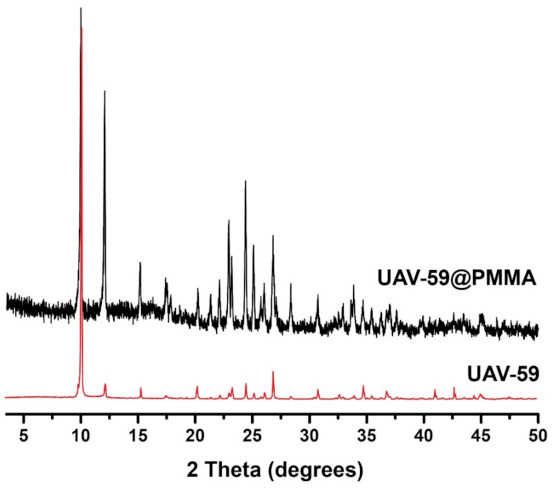
Comparison of the powder X-ray diffraction patterns of powdered UAV-59 and supported in the PMMA matrix

**Figure 3 molecules-26-02404-f003:**
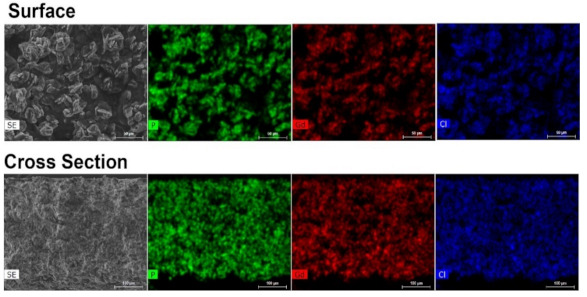
EDS mapping of UAV-59@PMMA of the membrane surface and of a cross section, showing a uniform distribution of the coordination polymer throughout the polymeric matrix.

**Figure 4 molecules-26-02404-f004:**
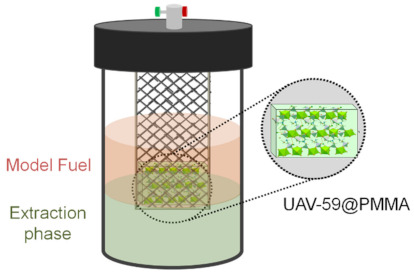
Schematic representation of the extraction and catalytic oxidative desulfurization process and the catalytic membrane reactor.

**Figure 5 molecules-26-02404-f005:**
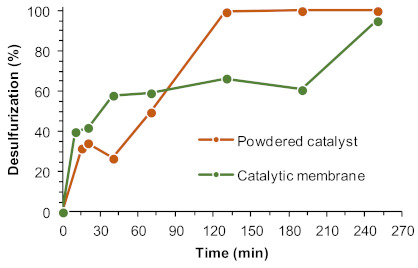
Desulfurization of a multicomponent model diesel catalyzed by the UAV-59 catalyst as a powder and as the UAV-59@PMMA membrane (both catalysts containing 0.04 mmol of Gd^3+^), using H_2_O_2_ as oxidant (0.64 mmol) and [BMIM]PF_6_ as the extraction solvent (1:1 diesel:solvent), at 70 °C. The lines connecting experimental points are only for guidance.

**Figure 6 molecules-26-02404-f006:**
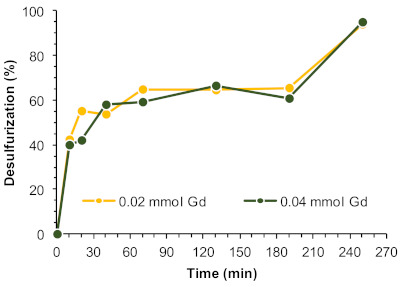
Desulfurization of a multicomponent model diesel catalyzed by different amounts of active centers, i.e., UAV-59@PMMA membrane containing 0.02 or 0.04 mmol of Gd^3+^, using H_2_O_2_ as oxidant (0.64 mmol) and [BMIM]PF_6_ as the extraction solvent (1:1 diesel:solvent), at 70 °C. The lines connecting experimental points are only for guidance.

**Figure 7 molecules-26-02404-f007:**
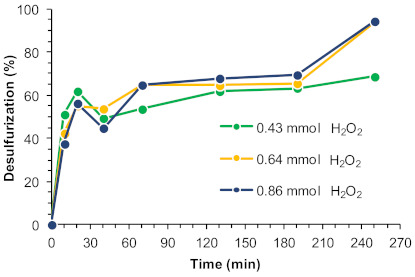
Desulfurization of a multicomponent model diesel (2000 ppm in sulfur) catalyzed by the UAV-59@PMMA membrane (containing 0.02 mmol Gd^3+^), using different amounts of H_2_O_2_ as oxidant (0.86, 0.64, and 0.43 mmol) and [BMIM]PF_6_ as extraction solvent (1:1 diesel:solvent), at 70 °C.

**Figure 8 molecules-26-02404-f008:**
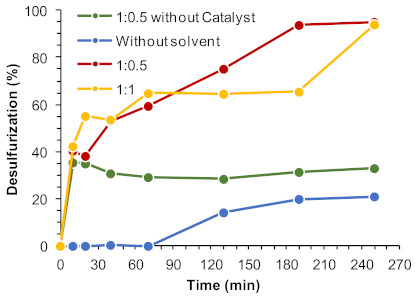
Desulfurization of a multicomponent model diesel (2000 ppm S) catalyzed by UAV-59@PMMA membrane (containing 0.02 mmol Gd), using 0.64 mmol of H_2_O_2_ as oxidant, different volume of extraction solvent (1:1 and 1:0.5 diesel/[BMIM]PF_6_) and for a solvent-free system, at 70 °C.

**Figure 9 molecules-26-02404-f009:**
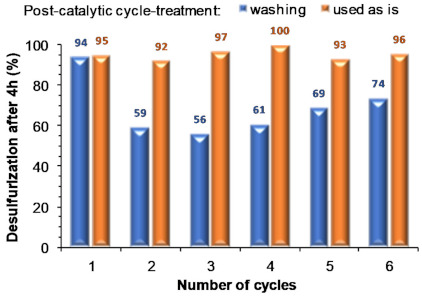
Desulfurization results obtained (after 4 h) for consecutive recycling ECODS cycles, catalyzed by the UAV-59@PMMA membrane (containing 0.02 mmol Gd), using 0.64 mmol of H_2_O_2_ oxidant, 1:0.5 model diesel:[BMIM]PF_6_, at 70 °C.

**Figure 10 molecules-26-02404-f010:**
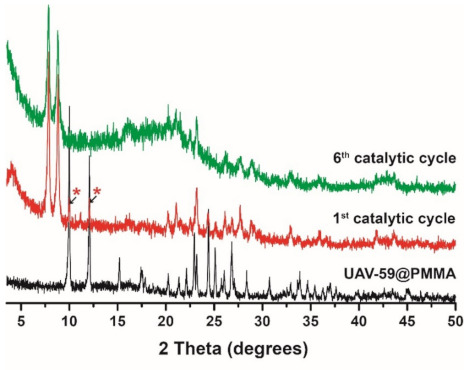
Powder X-ray diffraction of UAV-59@PMMA after the first and sixth catalytic cycles. The asterisks indicate the two main reflections from the parentUAV-59 material still present after the first catalytic cycle.

**Figure 11 molecules-26-02404-f011:**
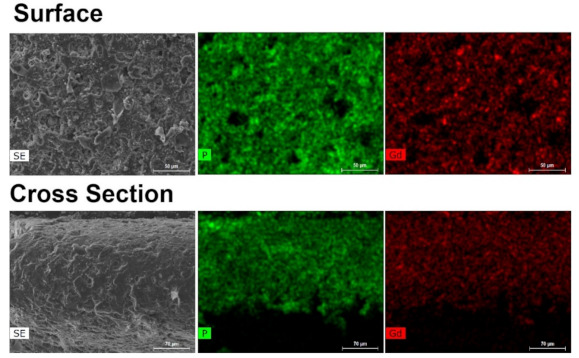
EDS mapping of UAV-59@PMMA at the membrane surface and a cross section after the catalytic studies.

## Data Availability

More data can be obtained by request from the authors.

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
