# Peer review of "Membrane-Supported Layered Coordination Polymer as an Advanced Sustainable Catalyst for Desulfurization"

_molecules, 2021, doi:10.3390/molecules26092404_

Round 1

Reviewer 1 Report

The manuscript contains very timely and valuable information about the abatement of sulfur containing compounds from a model diesel stream. It contains information how a catalytic membrane based technology can be used in this difficult technology/study area. Besides raw oil derived fuels, hydrocarbon streams originating from bio-based resources, e.g. lignocellulosic streams, can contain sulfur which may disturb catalysts used in biorefinary production processes.

The idea and its novelty seem very clear and timely. The manuscript, however, studies very many different things even though the main study area, i.e. the preparation and characterization of the catalytic membrane and its testing would have needed even more attention. The authors have also studied the deactivation and even sustainability of the new technology in a  broad context. Hopefully the authors will go in the becoming articles more deep down into the hybrid materials development as well as their selectivity, activity and stability issues.

Please, find below some comments (no all) to the text

Line 18: 'the' needs to be replaced by 'to'

Line 19: the definite articles are missing

Line 21: an uniform - the article needs to be 'a'

Line 26: 'that d' needs to be 'the'

Line 64: 'Other ' needs to be changed to 'another'

Line 89: 'influence in' needs to be changed to 'influence on'  and 'by way' to 'via'

Line 97: 'could not achieved' to 'could not be achieved'

Line 99: 'could considered' to 'could be considered'

Line 114: ...is the double than the same...???

Lines 118-120: change the sentence, it is not very clearly presented

Line 254: 'at different conditions' to 'in different conditions'

Line 330: 'preparation, .ca. 600 mg' to 'preparation, ca. 600 mg'

Line 332: 'and stirred' to 'which was stirred'

Line 363: 'catalytic active' to 'catalytically active'

Line 364: rewrite this line, i.e. in to...

In LIne 54, the word 'sustainable' should be considered. What does this mean - sustainability with economic, environmental and social dimensions? The word 'sustainable' is often used too loosely and its use should be based on assessment. 

Author Response

 Reviewer 1

The manuscript contains very timely and valuable information about the abatement of sulfur containing compounds from a model diesel stream. It contains information how a catalytic membrane based technology can be used in this difficult technology/study area. Besides raw oil derived fuels, hydrocarbon streams originating from bio-based resources, e.g. lignocellulosic streams, can contain sulfur which may disturb catalysts used in biorefinary production processes.

The idea and its novelty seem very clear and timely. The manuscript, however, studies very many different things even though the main study area, i.e. the preparation and characterization of the catalytic membrane and its testing would have needed even more attention. The authors have also studied the deactivation and even sustainability of the new technology in a  broad context. Hopefully the authors will go in the becoming articles more deep down into the hybrid materials development as well as their selectivity, activity and stability issues.

Please, find below some comments (no all) to the text

Line 18: 'the' needs to be replaced by 'to'

Line 19: the definite articles are missing

Line 122: an uniform - the article needs to be 'a'

Line 126: 'that d' needs to be 'the'

Line 165: 'Other ' needs to be changed to 'another'

Line 191: 'influence in' needs to be changed to 'influence on'  and 'by way' to 'via'

Line 199: 'could not achieved' to 'could not be achieved'

Line 201: 'could considered' to 'could be considered'

Line 217: ...is the double than the same...???

Lines 220-222: change the sentence, it is not very clearly presented

Line 258: 'at different conditions' to 'in different conditions'

Line 334: 'preparation, .ca. 600 mg' to 'preparation, ca. 600 mg'

Line 336: 'and stirred' to 'which was stirred'

Line 368: 'catalytic active' to 'catalytically active'

Line 369: rewrite this line, i.e. in to...

Response: All the comments mentioned by the reviewer were accepted and corrected as suggested. The sentence (lines 220-222) was rewrite “The percentage of desulfurization obtained without extraction solvent was 21% after 4 h of reaction which indicates that the oxidative catalytic reaction may occurs mostly in the [BMIM]PF6 phase.”

In LIne 154, the word 'sustainable' should be considered. What does this mean - sustainability with economic, environmental and social dimensions? The word 'sustainable' is often used too loosely and its use should be based on assessment.

Response: The authors acknowledge the reviewer by this comment. The word sustainable used in this work has as purpose the eco-friendly character, i.e. the absence of harmfulness for the environment. The nature of the solvent and oxidant used, as well as the absence of catalyst waste (without loss), promoted by the desulfurization system here proposed using catalytic membrane presents to be a sustainable procedure with a minimum harmful impact to environment. The economic feature from the catalytic system is related to the amount and the cost the reactants and the experimental conditions used. The sustainable and economic aspects of the novel catalytic system here presented promote the viability of this for industrial application.

Reviewer 2 Report

In the present manuscript, authors have investigated the preparation of a catalytic membrane by impregnating the UAV-59 into a supporting membrane (PMMA), which can improve the facility of catalyst handling and avoid catalysts’ deactivation. Various parameters are optimized to increase the desulfurization performance, stability and recyclability of the catalytic membrane. Although application of the PMMA catalytic membrane in oxidative desulfurization is relatively new, several serious issues should become clear, and these results do not provide more novel scientific views. Therefore, the quality of the paper needs to be carefully improved before reconsideration for publication in Molecules. I suggest that the acceptance of this paper needs to be carefully considered. There are a number of major issues that authors should carefully address in a revised version of the manuscript, as detailed below.

Major issues:

  1. In page 2 line 96, the optimal PMMA-to-UAV-59 ratio is, which is short of experimental validation. The specific shape of the pocket is not shown. Why not consider optimizing the casting method to make it better?
  2. In page 6 line 176, UAV-59@PMMA membrane incorporating 0.02 mmol of Gd3+ was selected after optimization. But the comparative experiment with powdered catalyst UAV-59 was carried using 0.04 mmol of Gd3+. It is necessary to provide two kinds of catalyst’s performance under the condition of 0.02 mmol of Gd3+.
  3. In page 6 line 195. Different of oxidant amount have no obvious effect. The same induction period was observed up to 3 h of reaction using 100 and 75 µL oxidant. Can the induction period be shortened by increasing the amount of oxidant?
  4. Different ECODS systems were investigated by using different model diesel to [BMIM]PF6 However, it is difficult to reflect its advantages due to the lack of desulfurization performance of powdered catalyst under optimized conditions. Advantages compared with traditional catalysts must be well documented and presented.
  5. Graph drawing is not intuitive enough which makes it hard to follow lines of reasoning and explanations.

Minor issues:

  1. The English language level is not adequate for publication at this stage. Though often understandable, most of the manuscript contains awkward uses of terms and construction of sentences, which makes it hard to follow lines of reasoning and explanations.
  2. Please clarify how the modification of the crystal morphology can be obtained.
  3. Why the authors choose the PMMA membrane in desulfurization experiments rather than other catalytic membrane? Please explain its superiorities.

Author Response

Point-by point answers to Reviewers

The authors acknowledge the reviewers’ helpful and relevant comments. They have received careful consideration on our part and have resulted in the introduction of some revisions in the manuscript that are highlighted green to reviewer 2,. Our responses are as follows:

 Reviewer 2

In the present manuscript, authors have investigated the preparation of a catalytic membrane by impregnating the UAV-59 into a supporting membrane (PMMA), which can improve the facility of catalyst handling and avoid catalysts' deactivation. Various parameters are optimized to increase the desulfurization performance, stability and recyclability of the catalytic membrane. Although application of the PMMA catalytic membrane in oxidative desulfurization is relatively new, several serious issues should become clear, and these results do not provide more novel scientific views. Therefore, the quality of the paper needs to be carefully improved before reconsideration for publication in Molecules. I suggest that the acceptance of this paper needs to be carefully considered. There are a number of major issues that authors should carefully address in a revised version of the manuscript, as detailed below.

Major issues:

  • In page 2 line 96, the optimal PMMA-to-UAV-59 ratio is, which is short of experimental validation. The specific shape of the pocket is not shown. Why not consider optimizing the casting method to make it better?

Response: Different PMMA-to-UAV-59 ratios were investigated prior to the preparation of the manuscript. As in any research work, these preliminary results were included in the manuscript. Nevertheless, the various studies performed (alongside with extensive structural characterization) led to a number of important conclusions which are nevertheless detailed in the experimental section: higher amounts of UAV-59 with respect to PMMA led to a lower mechanical stability of the composites, with the membrane breaking apart very easily; lower amounts of UAV-59 led to a lower homogeneity of the composite. Thus, we believe that the reviewer did not fully understand our explanation present in the manuscript. The pocket mentioned in his/her assessment concerns aggregated powdered material in different zones of the composite membrane, which correspond to regions with a higher UAV-59 concentration. As stated, these types of membranes were discarded and for the present study we selected the optimal PMMA-to-UAV-59 ratio that avoided all these problems.

Regarding the casting method, we agree with the reviewer. In fact, the optimization of the casting method (for example, using more sophisticated techniques such as spin coating) is a current interest which is not fully developed and implemented in our laboratories. We agree that this is an important topic to be investigated in the future.  

2)  2. In page 6 line 176, UAV-59@PMMA membrane incorporating 0.02 mmol of Gd3+ was selected after optimization. But the comparative experiment with powdered catalyst UAV-59 was carried using 0.04 mmol of Gd3+. It is necessary to provide two kinds of catalyst’s performance under the condition of 0.02 mmol of Gd3+.

Response: The authors would like to clarify this point. Before the optimization studies using catalytic membrane, the performance of this was compared to the powered UAV-59. Our research group, published recently the application of the UAV-59 heterogeneous catalyst in ODS ([33] Mirante et al., Catalysts 2020, 10) and in this work the optimized amount of powered catalyst was 0.04 mmol Gd3+. Therefore, the initial catalytic comparison between powdered and membrane was performed using 0.04 mmol (Figure 5). Later, an optimization of the amount of catalytic membrane was studied and the same catalytic efficiency was found using a piece of membrane containing 0.04 mmol and half the area of the same membrane, containing 0.02 mmol. Therefore, the further work was performed using 0.02 mmol of Gd3+.

3) In page 6 line 195. Different of oxidant amount have no obvious effect. The same induction period was observed up to 3 h of reaction using 100 and 75 µL oxidant. Can the induction period be shortened by increasing the amount of oxidant?

Response: The authors acknowledge the reviewer for this question. An additional experiment using 200 uL of H2O2 was performed: however, the desulfurization efficiency found was still similar to the previous obtained using 75 and 100 uL of oxidant. 

4) Different ECODS systems were investigated by using different model diesel to [BMIM]PF6 However, it is difficult to reflect its advantages due to the lack of desulfurization performance of powdered catalyst under optimized conditions. Advantages compared with traditional catalysts must be well documented and presented.

Response: The comparison of catalytic activity between the powdered and the membrane is presented in Figure 5 under the same experimental conditions and using the same model diesel. The results demonstrated that the powdered catalyst did not showed a longer induction period as the membrane catalyst. This discussion is already presented in the manuscript between lines 161 and 166. At this time the conditions were not the optimized for the membrane catalytic system. Under optimized conditions, i.e. 0.64 mmol H2O2, 0.02 mmol Gd and 1:0.5 model diesel/[BMIM]PF6, near complete desulfurization was achieved after 3 h, instead of 4 h using 1:1 model diesel/[BMIM]PF6.

       In our previous work (reference [33]), we used the powdered catalyst under the 1:0.5 model diesel/[BMIM]PF6 system (in electronic supporting information) and at this time near complete desulfurization was achieved after only 1 h. These results demonstrated that the powdered catalyst can achieve complete desulfurization in shorter time using a similar model diesel containing 1-BT, DBT, 4-MDBT and 4,6-DMDBT. To better clarify this point, the sentence presented in lines 367-368 was corrected and highlighted at green. 

The advantages of using a membrane catalyst instead of a powdered are mainly associated to the absence of catalyst mass loss and also the extra laborious catalyst separation and recovery procedures (presented between lines 15 and 18, and again 125 and 130).

5) Graph drawing is not intuitive enough which makes it hard to follow lines of reasoning and explanations.

Response: The authors acknowledge the reviewer for this. Some modifications were introduced to turn the graphical abstract more intuitive.

 Minor issues:

  1. The English language level is not adequate for publication at this stage. Though often understandable, most of the manuscript contains awkward uses of terms and construction of sentences, which makes it hard to follow lines of reasoning and explanations.

Response: The manuscript was read carefully and sentences were slightly modified in order to improve the text.

  1. Please clarify how the modification of the crystal morphology can be obtained.

Response: In this section of the manuscript we aimed to demonstrate that after the catalytic reaction the crystallinity of the materials tends to decrease, with crystallites breaking apart, with the concomitant modification of plate-like morphology observed for the parent material. This more detailed explanation was included in the revised manuscript. (line 285, p.9)

  1. Why the authors choose the PMMA membrane in desulfurization experiments rather than other catalytic membrane? Please explain its superiorities.

Response: In this report we chose a polymer capable of supporting the CP in its powdered form, which was also catalytic inert. Under these conditions we were able to compare the catalytic activity of the CP used in its powdered form and while supported and uniformly dispersed. PMMA was selected because it is one of the most used polymers in different industries, while it is thermally stable in the conditions, light weight and resistant to impact. This explanation was introduced in the revised manuscript. (line 79, p.2)

Round 2

Reviewer 2 Report

It is OK for now.